# *Escherichia coli* tRNA 2-Selenouridine Synthase (SelU): Elucidation of Substrate Specificity to Understand the Role of *S*-Geranyl-tRNA in the Conversion of 2-Thio- into 2-Selenouridines in Bacterial tRNA

**DOI:** 10.3390/cells11091522

**Published:** 2022-05-02

**Authors:** Patrycja Szczupak, Malgorzata Sierant, Ewelina Wielgus, Ewa Radzikowska-Cieciura, Katarzyna Kulik, Agnieszka Krakowiak, Paulina Kuwerska, Grazyna Leszczynska, Barbara Nawrot

**Affiliations:** 1Centre of Molecular and Macromolecular Studies, Polish Academy of Sciences, Sienkiewicza 112, 90-363 Lodz, Poland; pkomar@cbmm.lodz.pl (P.S.); ms@cbmm.lodz.pl (E.W.); eradziko@cbmm.lodz.pl (E.R.-C.); kpieta@cbmm.lodz.pl (K.K.); akrakow@cbmm.lodz.pl (A.K.); bnawrot@cbmm.lodz.pl (B.N.); 2Institute of Organic Chemistry, Lodz University of Technology, Zeromskiego 116, 90-924 Lodz, Poland; paulina.kuwerska@dokt.p.lodz.pl (P.K.); grazyna.leszczynska@p.lodz.pl (G.L.)

**Keywords:** tRNA 2-selenouridine synthase, MBP–SelU fusion protein, modified nucleoside, 2-thiouridine, *S*-geranyl 2-thiouridine, 2-selenouridine

## Abstract

The bacterial enzyme tRNA 2-selenouridine synthase (SelU) is responsible for the conversion of 5-substituted 2-thiouridine (R5S2U), present in the anticodon of some bacterial tRNAs, into 5-substituted 2-selenouridine (R5Se2U). We have already demonstrated using synthetic RNAs that transformation S2U→Se2U is a two-step process, in which the S2U-RNA is geranylated and the resulting geS2U-RNA is selenated. Currently, the question is how SelU recognizes its substrates and what the cellular pathway of R5S2U→R5Se2U conversion is in natural tRNA. In the study presented here, we characterized the SelU substrate requirements, identified SelU-associated tRNAs and their specific modifications in the wobble position. Finally, we explained the sequence of steps in the selenation of tRNA. The S2U position within the RNA chain, the flanking sequence of the modification, and the length of the RNA substrate, all have a key influence on the recognition by SelU. MST data on the affinity of SelU to individual RNAs confirmed the presumed process. SelU binds the R5S2U-tRNA and then catalyzes its geranylation to the R5geS2U-tRNA, which remains bound to the enzyme and is selenated in the next step of the transformation. Finally, the R5Se2U-tRNA leaves the enzyme and participates in the translation process. The enzyme does not directly catalyze the R5S2U-tRNA selenation and the R5geS2U-tRNA is the intermediate product in the linear sequence of reactions.

## 1. Introduction

Transfer RNAs (tRNAs) are present in three domains of life as a universal component of cellular protein synthesis machinery. To become fully functional, tRNAs, which were originally transcribed in the form of longer precursors, undergo a well-defined multistep maturation process, in which an important step that influences tRNA structure and functionality is the incorporation of numerous post-transcriptional chemical modifications. The four canonical RNA bases (A, C, G, and U) are modified in different ways by specific enzymes [1,2,3,4].

Natural tRNAs contain a highest proportion of modified nucleosides compared to other RNA types [5,6]. The post-transcriptional modifications in tRNA can be divided into two groups based on their complexity. The first group of modifications is introduced by a single enzymatic reaction, such as methylation, and is found at numerous locations within the tRNA core region, in which tertiary interactions between the D and T arms stabilize the three-dimensional structure. The second group includes complex modifications whose syntheses require the sequential activity of multiple enzymes. These hyper-modifications are mainly found in the anticodon loop, in which they maintain the structure as a prerequisite for efficient translation [7]. The anticodon stem-loop (ASL) is the most frequently modified part of the tRNA molecule, particularly nucleosides in the wobble position (which is also referred to as the 1st position in the anticodon or the 34th position in the tRNA chain), which interact with the third base of the codon in the mRNA. The second commonly modified position in the ASL is the 37th position, which is located directly after the last base of the anticodon. The modification of the nucleosides in these positions play a crucial role in the extension or restriction of the decoding properties of a particular tRNA molecule [8]. Positions outside the ASL usually play a more structural role, but they have also been shown to influence decoding properties [7,9].

Hypermodified uridines that contain sulfur or selenium atom are widely distributed in a variety of organisms and have been identified in the wobble position of the anticodon sequence of tRNA^Lys^, tRNA^Glu^, and tRNA^Gln^. Moreover, in 2012, Dumelin et al. discovered a new type of modification with a hydrophobic geranyl group conjugated to the sulfur atom of the 2-thiouridine nucleoside in bacterial tRNAs [10]. Figure 1 shows the structure of 5-methylaminomethyl-2-thiouridine (mnm5S2U) and its counterparts: 5-methylaminomethyl-*S*-geranyl-2-thiouridine (mnm5geS2U) and 5-methylaminomethyl-2-selenouridine (mnm5Se2U). These counterparts are specific to the bacterial tRNAs for lysine (tRNA^Lys^) and glutamic acid (tRNA^Glu^). The bacterial tRNA specific to glutamine (tRNA^Gln^) contains 5-carboxymethylaminomethyl-2-thiouridine (cmnm5S2U) and its counterparts: 5-carboxymethylaminomethyl-*S*-geranyl-2-thiouridine (cmnm5geS2U) and 5-carboxymethylaminomethyl-2-selenouridine (cmnm5Se2U) [5,6,10,11].

Sulfur and selenium modifications in the tRNA wobble uridines play important role in the precise reading of genetic information and the tuning of the protein synthesis [12,13]. We and other researchers have attempted to explain why nature has introduced such a complex system of modifications into the wobble position of bacterial tRNAs [14,15,16,17,18,19]. Uridines in the RNA chain preferentially recognize the A-complement through Watson–Crick interactions and, with a lower affinity, the G-complement through the wobble hydrogen bonding pattern [16]. The introduction of sulfur or selenium atoms into the C2 position of uridine increases the thermodynamic stability of RNA duplexes that contain S2U-A or Se2U-A base pairs and restricts the formation of S2U-G or Se2U-G base pairs [17]. However, the situation changes when the side chain is present in the C5 position of 2-thiouridines and 2-selenouridines [14]. Under physiological conditions (pH 7.4), R5Se2Us preferentially adopt the zwitterionic form (ZI, about 90%) with the positive charge on the aminoalkyl side chain and the negative charge on the Se2-N3-O4 edge. The tRNA anticodons with the R5Se2Us wobble can also read the synonymous 5′-NNG-3′ codons in mRNA in contrast to their 2-oxo precursors, which preferentially read the 5′-NNA-3′ codons [14], and to their 2-thio precursors, which recognize both synonymous codons depending on the character of the substituent in the C5 position. Moreover, the presence of sulfur and selenium in biological entities, such as proteins or nucleic acids, can be expected to protect cellular elements from oxidative damage. Selenium-containing compounds are superior ROS scavengers because, unlike sulfur analogs, they can react reversibly with oxidizing species [18,19].

The enzyme that is responsible for the subsequent conversion of S2U→Se2U is the subject of our current research [11,20]. The enzymatic activity in extracts of *Salmonella enterica* serovar *Typhimurium*, which catalyzes the conversion of mnm5S2U into mnm5Se2U in the presence of selenide and ATP, was first described by Verez et al. in 1992 [21]. Two years later, Veres and Stadtman isolated and purified the enzyme that is responsible for this conversion and introduced the name tRNA 2-selenouridine synthase (SelU) [22]. In 2012, Dumelin et al. demonstrated that SelU is responsible for the introduction of the geranyl group into two types of R5-substituted thio-nucleosides (mnm5S2U and cmnm5S2U) in bacterial tRNAs [10]. The tRNA 2-selenouridine synthase (SelU, MnmH, and YbbB) is a 41.1 kDa protein that contains a 364-amino acid chain, which is divided into two structural domains: an N-terminal domain with rhodanese homology, with a -Cys-X-X-Cys- active site, and a C-terminal P-loop domain, which contains a Walker A motif and an isoleucine–tRNA synthetase (IleS)-like helical region [23,24]. The P-loop domain is found in proteins that bind ATP or GTP [25,26]. The intact Walker A motif is required for the geranylation activity of the enzyme, which could mean it is also the binding site for geranyl pyrophosphate (GePP), a donor molecule in the geranyl transfer reaction in vitro [23]. This information was confirmed by the validation of molecular dynamics in our recently published study [27]. SelU synthase, under the natural conditions in the bacterial cells, contains a tightly bound specific tRNA fraction. Therefore, the purified protein has an unusual absorption spectrum with a maximum at 260 nm, as in nucleic acids, and no peak at 280 nm, which is characteristic of proteins. Wolfe et al. estimated that one protein molecule binds two tRNA molecules [24]. To date, the crystal structure of the SelU protein has not been determined. The putative 3D structure of the SelU protein was recently predicted based on its amino acid sequence using the AlphaFold v2.0 system. The structure is available for analysis on the AlphaFold Protein Structure Database website (https://alphafold.ebi.ac.uk/entry/P33667, accessed on 1 July 2021) [28].

Originally, the SelU enzyme was thought to be responsible for the direct conversion of R5S2U→R5Se2U and, additionally, for the *S*-geranylation of the R5S2U-tRNA as a second independent modification pathway. Dumelin et al. claimed that the *S*-geranylation of tRNA is an alternative to the selenation process, which is prolonged at low selenium concentrations [10]. For several years, the selenation and geranylation of the R5S2U-tRNA were thought to occur independently and in parallel. Since these two reactions (S2U→Se2U and S2U→geS2U) are mechanistically distinct, we attempted to elucidate the reaction pathway, first using chemical conversions and then using enzymatic conversions [11,20,29]. In 2018, we demonstrated using 17-mer model ASL–RNAs that the conversion of the S2U-RNA into the Se2U-RNA occurs via the geS2U-RNA intermediate product, which corresponds to the two subsequent reactions cycle: S2U-RNA→geS2U-RNA→Se2U-RNA [11].

In previous studies, we used the wild type SelU protein, whose gene was isolated from the *E. coli* RNA through reverse transcription followed by PCR with specific primers, and cloned it into the DNA expression plasmid pET28c, which also encoded a short His_6_ tag. [11,20]. The recombinant SelU–His_6_ protein was not stable under the conditions applied, so the yield of geranylation was only ~40% for a freshly prepared sample and decreased to ~10% for the protein stored at −70 °C for several weeks [11,20]. This enzyme efficiency was not sufficient for us to explain the properties and differences of the interactions with the substrate that were studied. Therefore, we decided to change the system and constructed an MBP–SelU fusion protein with an MBP tag (42 kDa) at the N-terminus of the SelU protein. In our case, the MBP tag acted as a stabilizing factor in the MBP–SelU fusion protein [30]. The MBP-modified SelU retained the substrate preferences, tRNA-binding properties, and catalytic ability of SelU synthase but showed much higher stability and activity in the specific reactions: geranylation (>90%) and selenation (100%). Thanks to these parameters of the MBP–SelU fusion protein, we obtained a molecular tool that can be used to characterize and evaluate the properties of SelU synthase.

## 2. Materials and Methods

### 2.1. The Chemical Synthesis of Modified Nucleosiede Standards

The chemical syntheses of an R5-substituted 2-oxo-, 2-thio-, S-geranyl-2-thio-, and 2-selenouridines was performed at the Lodz University of Technology and we used the standards for LC-MS analysis. The procedures for the synthesis of nm5U, mnm5U, cmnm5U, S2U, nm5S2U, mnm5S2U, cmnm5S2U, Se2U, mnm5Se2U, cmnm5Se2U, geS2U, mnm5geS2U, and cmnm5geS2U have been previously described [14,17,31,32,33]. The procedure for the synthesis of the nm5geS2U and nm5Se2U standards is presented in the Synthetic Procedures in the Appendix A.

### 2.2. The Chemical Synthesis of S2U and geS2U Phosphoramidites and Model RNAs Oligonucleotides

The chemical syntheses of the S2U and geS2U phosphoramidites and the model RNA oligonucleotides were performed in-house according to previously described procedures [11,20,34]. The detailed procedures are presented in the Synthetic Procedures in the Appendix A. The list of prepared RNA models, their sequences, and the mass spectrometry analysis data are presented in Appendix A.

### 2.3. The Chemical Synthesis of Selenophosphate

The synthesis of selenophosphate SePO_3_^3−^ was performed according to the procedure previously described [20,35,36]. The detailed procedure is presented in the Synthetic Procedures in the Appendix A.

### 2.4. Protein Procedures

The detailed procedure for the synthesis and purification of the MBP–SelU fusion protein is presented in the Protein Procedures in the Appendix A. The SDS-PAGE validation of the protein overexpression level and the purification of MBP–SelU are presented in Appendix A.

### 2.5. Geranylation Reaction 

The geranylation reaction conditions are described in Appendix A: 21 µg (0.249 nmol) of purified MBP–SelU was incubated with 6.11 µg (1.113 nmol) of the S2U-RNA in the presence of a 5-fold excess of geranyl pyrophosphate ammonium salt (GePP 5 × H_2_O, Axon Medchem, Reston, VA, USA) (5.56 nmol, 2.03 µg) in 100 μL of the reaction buffer (10 mM Tricine–KOH, pH 7.2, 10 mM MgCl_2_) at 37 °C for 1 h. The reaction products were analyzed using RP-HPLC (Shimadzu, Kyoto, Japan) with the Kinetex C-18 column, (5 µm, 100 A, and 250 × 4.60 mm) (Phenomenex, Warsaw, Poland) and buffers (A: 0.1 M CH_3_COONH_4_ and B: 0.1 M CH_3_COONH_4_/40% CH_3_CN). The reaction products were separated in a linear acetonitrile gradient: 0–5 min for 0% B, 5–21 min for 0–80% B, and 21–26 min for 80–0% B. The reaction products present in the collected fractions were identified by ESI-MS.

### 2.6. Selenation Reaction

The selenation reaction conditions are described in Appendix A: 21 µg (0.249 nmol) of purified MBP–SelU was incubated with 6.11 µg (1.113 nmol) of the geS2U-RNA in the presence of 18 equivalents of SePO_3_^3−^ (20 nmol, 3.16 µg) in 100 μL of the reaction buffer (10 mM Tricine–KOH, pH 7.2, 10 mM MgCl_2_), at 37 °C for 30 min, under anaerobic conditions. The Se2U-RNA product formation was monitored using RP-HPLC (Shimadzu, Kyoto, Japan) with the Kinetex C-18 column (5 µm, 100 A, and 250 × 4.60 mm), (Phenomenex, Warsaw, Poland) and buffers (A: 0.1 M CH_3_COONH_4_ and B: 0.1 M CH_3_COONH_4_ / 40% CH_3_CN). The reaction products were separated in a linear acetonitrile gradient: 0–5 min for 0% B, 5–21 min for 0–80% B, and 21–26 min for 80–0% B. The reaction products present in the collected fractions were identified by ESI-MS.

### 2.7. Kinetics of Geranylation and Selenation Reactions

The kinetic parameters of the S2U-RNA geranylation catalyzed by the MBP–SelU enzyme were determined in the reactions, in which the initial rates were maintained until approximately 10% of the S2U-RNA was converted into geS2U-RNA. The reactions were performed in the 200 µL reaction buffer containing 10 mM Tricine–KOH, pH 7.2, 10 mM MgCl_2_, with the constant GePP concentration (30 µM) and variable concentrations of the S2U-RNA oligonucleotide: 0, 0.625, 1.25, 2.5, 5, 7.5, 15, 20, and 25 µM. The enzyme was used in two concentrations: 8.3 pmol (0.041 µM) for lower concentrations of the oligonucleotide (0.625–7.5 μM) and 16.6 pmol (0.083 µM) for higher concentrations of the oligonucleotide (15–25 μM). The reaction products were separated using RP-HPLC (Shimadzu, Kyoto, Japan) with the Kinetex C-18 column (5 µm, 100 A, and 250 × 4.60 mm) (Phenomenex, Warsaw, Poland) in a linear acetonitrile gradient: 0–5 min for 0% B, 5–21 min for 0–80% B, and 21–26 min for 80–0% B (A: 0.1 M CH_3_COONH_4_ and B: 0.1 M CH_3_COONH_4_/40% CH_3_CN). Each reaction was repeated at least three times. The obtained results were analyzed using GraphPad Prism 4.0 software to plot the V(S) dependence curve, were V means reaction rate and S is the substrate concentration. The kinetic constants K_M_ and k_cat_ were determined.

To determine the kinetic parameters for the selenation of the geS2U-RNA catalyzed by the MBP–SelU enzyme, the initial rates were maintained until approximately 10% of the geS2U-RNA was converted into Se2U-RNA. The reactions were performed in the 200 µL reaction buffer containing 10 mM Tricine–KOH, pH 7.2, 10 mM MgCl_2_, with a constant concentration of SePO_3_^3−^ (30 µM) and variable concentrations of the geS2U-RNA oligonucleotide: 0, 0.3125, 0.625, 1.25, 2.5, and 5 μM. The enzyme concentration varied from 0.0386 pmol (0.193 nM) for the lower concentrations of the geS2U-RNA to 0.622 pmol (3.11 nM) for the higher concentrations of oligonucleotide. The reaction products were separated using RP-HPLC (Shimadzu, Kyoto, Japan) with a Kinetex C-18 column (5 µm, 100 A, and 250 × 4.60 mm) (Phenomenex, Warsaw, Poland) in the linear acetonitrile gradient: 0–5 min for 0% B, 5–21 min for 0–80% B, and 21–26 min for 80–0% B (A: 0.1 M CH_3_COONH_4_ and B: 0.1 M CH_3_COONH_4_/40% CH_3_CN). Each reaction was repeated three times. The obtained results were analyzed using GraphPad Prism 4.0 software to plot the V(S) dependence curve and to determine the kinetic constants K_M_ and k_cat_.

### 2.8. Microscale Thermophoresis (MST) Experiments

The relative binding affinity between the MBP–SelU protein and the Cy3-labeled ASL–RNA oligonucleotides, containing the unmodified uridine (U-) or C2 modified uridine (S2U-, geS2U- or Se2U-, respectively) in the position that mimicked the wobble position in tRNA (sequences are given in Appendix A), was investigated using microscale thermophoresis (MST) with the Monolith NT.115 system (NanoTemper Technologies, München, Germany). The 20 µL samples prepared for the MST measurements contained fluorescently labeled RNA oligonucleotide (target) at the constant concentration and the unlabeled protein (ligand) at sixteen serial dilutions (1:1). The samples were prepared according to the protocol described in the User Starting Guide for the Monolith NT.115 (NanoTemper Technologies, München, Germany). The working concentration of the Cy3-RNA oligonucleotides was chosen based on the fluorescence intensity, which should range from 200 to 1500 counts. The type of capillaries used for the MST measurements was chosen based on the shape of the fluorescence peaks, which were tested through Capillary Scan before the start of the experiments. The MBP–SelU ligand (452 µM) was subjected to the series of dilutions (from 226 µM to 0.0069 µM) in 2-times concentrated assay buffer (20 mM Tricine, pH 7.2, 20 mM MgCl_2_, 0.1% Tween-20) and mixed with the Cy3-RNA and incubated for 30 min at room temperature. Then, the mixtures were centrifuged (15,000 rpm for 5 min) and the samples were loaded into the Monolith NT.115 Standard Treated Capillary (Nanotemper Technologies, München, Germany) and measured with 40% MST power, 20% LED excitation power (excitation type: green) at a constant temperature of 22 °C. The measurements were supported by the MO Control v2.2.4 software (NanoTemper Technologies, München, Germany). The experiments were repeated at least three times for each pair of interacting molecules: MBP–SelU:U-RNA, MBP–SelU:S2U-RNA, MBP–SelU:geS2U-RNA, and MBP–SelU:Se2U-RNA. As a control, the interactions between the MBP protein and the tested oligonucleotides were examined. The initial fluorescence signals were analyzed using MO Affinity Analysis v2.2.4 software (NanoTemper Technologies, München, Germany) and the interaction affinity and dissociation constant (K_d_) were determined for each target–ligand pair using the K_d_ fitting model. The protein–RNA binding specificity was confirmed by the SD assay based on the denaturation of the protein in the samples (capillaries: 1, 2, and 3 vs. 14, 15, and 16) using a mixture of 4% SDS and 40 mM DTT in combination with heating to 95 °C, followed by the measurement of fluorescence intensity using a Monolith NT.115 instrument. When the fluorescence intensities of the samples were identical after denaturation, it could be concluded that the fluorescence changes observed before the denaturation were triggered by a binding event.

### 2.9. Isolation of tRNA Associated with MBP-SelU Protein

The fraction of the purified MBP–SelU protein (purity ~99%) was dissolved in the 400 µL of storage buffer (20 mM Tris-HCl, pH 7.4, 25 mM NaCl) and mixed with an equal volume of phenol solution, equilibrated with 10 mM Tris-HCl, pH 7.4 (Sigma-Aldrich, Poznan, Poland), vortexed and centrifuged at maximum speed for 15 min. Then, the clear aqueous phase was transferred into a new eppendorf tube, treated with 400 µL of a phenol–chloroform mixture (1:1), and centrifuged as above. The aqueous phase was transferred into the new tube and washed twice with chloroform (400 µL) to remove the phenol residues. The nucleic acid present in the aqueous phase was precipitated with 2.5 volumes of 99% ethanol with the addition of 0.1 volume of 3 M sodium acetate, pH 5.0. The obtained pellet was washed with 70% ethanol and air dried. The same procedure was repeated for the isolation of tRNA from the pure MBP protein to check whether the MBP tag bound the nucleic acids (control isolation). The amount of isolated tRNA was assessed spectrophotometrically after measuring the Abs_260_.

For the analysis of the full-length tRNAs associated with the MBP–SelU, the protein sample (200 µg) in the storage buffer was thermally denatured (5 min at 95 °C) and centrifuged at maximum speed for 15 min at 4 °C. The supernatant was collected and analyzed using UPLC-PDA-ESI(-)-MS.

### 2.10. Isolation of tRNA^Lys^, tRNA^Glu^ and tRNA^Gln^ from E. coli^ΔSelU^

The *Escherichia coli*, strain BW25113 with the SelU knockout (*E. coli* Genetic Stock Center Yale College, New Haven, CT, USA) was cultured in 12 L of LB medium (with 1% inoculum from the overnight culture) at 37 °C for several hours (~4–5 h) until the Abs_600_ reached 0.6–0.8 OD. Then, the culture was centrifuged (4000 rpm for 30 min at 4 °C) and the cells were lysed in TriReagent (ThermoFisher Sci., Waltham, MA, USA). The total cellular RNA was isolated according to the manufacturer’s procedure for TriReagent. The individual tRNAs were isolated from the total cellular RNA mixture using a specific DNA probes labeled with biotin at the 5′ end. The probe specific for tRNA^Lys^ was 5′-biotin-TGCGACCAATTGATTAAAAGTCAACTGCTC-3′. The probe specific for tRNA^Glu^ was 5′-biotin-CCTGTTACCGCCGTGAAAGGGCGGTGTCC-3′. The probespecific for tRNA^Gln^ was 5′-biotin-AGGGAATGCCGGTATCAAAAACCGGTGCCT-3′. They were all bound to Streptavidin Agarose resin (ThermoFisher Sci.). The total cellular RNA resuspended in 10 mM Tris-HCl, pH 7.4, and 150 mM NaCl buffer was added to the agarose beads with bound the appropriate probe and incubated at 85 °C for 5 min. Then, the beads were cooled to room temperature for 10 min and incubated overnight at 4 °C with vigorous mixing (2000 rpm). The next day, the samples were centrifuged (3000 rpm for 2 min) to remove unbound RNA and washed with 10 mM Tris-HCl pH 7.4 buffer until the Abs_260_ of the solution reached zero. The bound tRNA was eluted by the successive addition of 200 μL of deionized water, incubation at 85 °C for 3 min, and centrifugation. The harvested tRNA-containing solution was evaporated in a vacuum centrifuge (Savant). The obtained full-length tRNAs were subjected to UPLC-PDA-ESI(-)-MS analysis and, in the final stage of the experiments, geranylation and selenation reactions.

### 2.11. The tRNA Nucleolytic Hydrolysis

Tthe natural tRNA sample (10 µg) was hydrolyzed into single nucleosides using a combination of two nucleases: Benzonase from *Serratia marcescens* (Sigma-Aldrich, Poznan, Poland), 20 units per reaction in 50 mM Tris-HCl, pH 8.0, 1 mM MgCl_2_ buffer for 4 h at 37 °C; Phosphodiesterase I from *Crotalus adamanteus* venom (Sigma-Aldrich), 0.8 units per reaction in 50 mM Tris-HCl.0, pH 8.0, and 20 mM of MgCl_2_ for 16 h at 37 °C; and finally Alkaline Phosphatase (EURx, Gdansk, Poland), 10 units per reaction in the manufacturer’s buffer (1 M diethanolamine, 10 mM p-nitrophenylophosphate, and 0.25 mM MgCl_2_, pH 9.8) for 1 h at 37 °C. Each sample after hydrolysis was filtered using a 10 000-MW cut-off spin filter (Merck, Poznan, Poland) and dried in a vacuum centrifuge. The obtained mixture of nucleosides was analyzed using UPLC-PDA-ESI(-)-HRMS.

### 2.12. The In Vitro Transformation of the R5S2U-tRNA into R5Se2U-tRNA

The natural (c)mnm5S2U-tRNAs (tRNA^Lys^, tRNA^Gln^, and tRNA^Glu^) isolated from the *E. coli*^ΔSelU^ were used to perform the geranylation and selenation reactions. For each type of tRNA, three reaction mixtures were prepared, each containing 5 µg of tRNA (0.2 nmol) in 10 mM Tricine-HCl buffer (pH 7.2) and 10 mM MgCl_2_. For the geranylation reaction (i), the R5S2U-tRNA was incubated with 5-fold molar excess (1 nmol) of the geranyl pyrophosphate (GePP) and MBP–SelU enzyme (2 µg = 0.0236 nmol), at 37 °C for 2 h. For the geranylation and subsequent selenation reactions (ii), the R5S2U-tRNA was incubated with 5-fold molar excess (1 nmol) of the geranyl pyrophosphate (GePP) and MBP–SelU enzyme (2 µg = 0.0236 nmol), at 37 °C for 2 h. After the geranylation step, the SePO_3_^3−^ (~4 nmol) and the additional portion of MBP–SelU (1 µg = 0.0118 nmol) were added into the mixture and the reaction was continued for the next 1 h at 37 °C. For the direct selenation reaction (iii), the R5S2U-tRNA was incubated with the MBP–SelU enzyme (3 µg and 0.0355 nmol, respectively) in the presence of the SePO_3_^3−^ (~4 nmol), at 37 °C for 1 h. The total amount of enzyme used in the second and third reactions (ii and iii) was 3 µg = 0.0355 nmol, which corresponded to approximately 0.8 µg (0.035 nmol) of the protein-bound tRNA fraction being added into the reaction mixture, according to our calculation of the amount of tRNA that was isolated from the pure protein. This amount of specific tRNAs that was present in the reaction was taken as background tRNA. All reactions were stopped by the thermal inactivation of the enzyme (95 °C for 3 min) and then the mixtures were centrifuged for the separation of the denatured protein pellets. The identification of the reaction products that were in the form of full-length tRNAs, was performed using the UPLC-PDA-ESI(-)-MS analysis or, after nucleolytic hydrolysis and dephosphorylation, using the UPLC-PDA-ESI(-)-HRMS analysis of the nucleosides that were derived from the tRNA products.

### 2.13. Ultra-Performance Liquid Chromatography Coupled with a Mass Spectrometry and Photodiode Array Detection

The LC-MS analysis was carried out on ACQUITY UPLC I Class chromatography system equipped with a photodiode array detector with a binary solvent manager (Waters Corp., Milford, MA, USA) coupled with an SYNAPT G2-Si mass spectrometer equipped with an electrospray source and quadrupole, Time-of-Flight mass analyzer (Waters Corp., Milford, MA, USA). 

The conditions for the nucleoside mixture analysis using the UPLC-PDA-ESI(-)-HRMS method: the ACQUITY HSS T3 1.8 μm column (100 × 2.1 mm, 1.7 µm) (Waters Corp., Milford, MA, USA) and HSS T3 guard column (2.1 × 5 mm, 1.8 µm), which was at 25 °C, were used for the chromatographic separation of the analyte. A gradient program was employed for the mobile phase, which combined solvent A (0.01% formic acid in water) and solvent B (0.01% formic acid in acetonitrile) as follows: 0% B (0–1.0 min), 0–0.2% B (1.0–2.4 min), 0.2–0.8% B (2.4–3.8 min), 0.8–1.8% B (3.8–5.2 min), 1.8–3.2% B (5.2–6.6 min), 3.2–5% B (6.6–10.0 min), 5–8% B (10.0–13.5 min), 8.0–30% B (13.5–15.5 min), 30–100% B (15.5–19.5 min), 100–100% B (19.5–20.5 min), 100–0% B (20.5–21.0 min), and 0% B (21.0–25.0 min). The flow rate was 0.2 mL/min and the injection volume was 5 μL. For the mass spectrometric detection, the electrospray source was operated in negative high-resolution mode at 30,000 FWHM, which resolved the power of the TOF analyzer. To ensure accurate mass measurements, the data were collected in centroid mode and the mass was corrected during acquisition using leucine encephalin solution as an external reference, Lock-Spray TM (Waters Corp., Milford, MA, USA), which generated a reference ion at *m*/*z* 554.2615 [M-H]^−^ in negative ESI mode. The optimized source parameters were: capillary voltage, 2.8 kV for negative mode; cone voltage, 20 V; source temperature, 110 °C; desolvation gas (nitrogen) flow rate, 600 L/h; temperature, 350 °C; nebulizer gas pressure, 6.5 bar. The mass spectrometer conditions were optimized using the direct infusion of the standard solution. The mass spectra were recorded over a range of *m*/*z* 100 to 1200. The PDA spectra were measured over the wavelength range of 210–400 nm in steps of 1.2 nm. The results of the measurements were processed using the MassLynx 4.1 software (Waters Corp., Milford, MA, USA), which was incorporated into the instrument.

The conditions for the full-length tRNA analysis using the UPLC-PDA-ESI(-)-MS method: the ACQUITY UPLC^®^ Oligonucleotides BEH C18 column (50 × 2.1 mm, 1.7 μm), which was maintained at 60 °C, was used for the chromatographic separation of analyte. A gradient program was employed for the mobile phase, which combined solvent A (15 mM of triethylamine and 400 mM of 1,1,1,3,3,3- hexafluoro-2-propanol in water) and solvent B (50% methanol and 50% solvent A, *v*/*v*) as follows: 42.5% B (0–1.0 min), 42.5–52% B (1.0–22.0 min), 52–52% B (22.0–24.0 min), 52–42.5% B (24.0–24.1 min), and 42.5–42.5% B (24.1–28 min). Water was used as the weak wash solvent and 10% methanol in water was used as the strong wash solvent. The flow rate was 0.2 mL/min and the injection volume was 10 μL. For the mass spectrometric detection, the electrospray source was operated in a negative resolution mode. The optimized source parameters were: capillary voltage, 2.7 kV; cone voltage, 40 V; desolvation gas flow rate, 600 L/h; temperature, 400 °C; nebulizer gas pressure, 6.5 bar; source temperature, 120 °C. The mass spectra were recorded over the range of *m*/*z* 500 to 2000. The mass spectrometer conditions were optimized using the direct infusion of the standard solution. The system was controlled using the MassLynx software v 4.1. The raw ESI mass spectra were deconvoluted to a zero-charge state mass using the MaxEnt1 algorithm. 

## 3. Results

### 3.1. Expression and Purification of tRNA 2-Selenouridine Synthase (SelU) Obtained in Fusion with Maltose-Binding Protein (MBP)

The pMAL, Protein Fusion and Purification System was chosen for the cloning and expression of the tRNA 2-selenouridine synthase (SelU) gene. The cloned gene was inserted downstream of the *malE* gene encoding maltose-binding protein (MBP) in the pMAL-c5x expression vector, resulting in the synthesis of the fusion protein with the N-terminal MBP tag. The recombinant fusion gene was overexpressed, according to the procedure described in the Protein Procedures, in the Appendix A. Appendix A shows the level of MBP–SelU expression in the total soluble bacterial proteins. The expression level of the recombinant SelU protein was low, regardless of the bacterial culture conditions or the use of different bacterial expression systems (data not shown). After the application of the two-step purification, the purity of the MBP–SelU samples that was estimated using densitometry was ~99% (Uvitec) (Appendix A). 

### 3.2. Analysis of the Catalytic Activity of MBP–SelU

The enzymatic activity of the MBP–SelU synthase in the geranylation and selenation reactions (Figure 1) was determined using ASL–RNA oligonucleotide substrates (17-mers) S2U(34)–RNA^Lys^ (5′-GUUGACUS2UUUAAUCAAC-3′) and geS2U(34)–RNA^Lys^ (5′-GUUGACUgeS2UUUAAUCAAC-3’), respectively. These RNAs had the same nucleotide sequence but differed in the single S2U or geS2U modification in the position that corresponded to the wobble site of the tRNA^Lys^ from *E. coli*. The detailed protocol of the reaction conditions is described in the Materials and Methods sectionand in Appendix A. After the geranylation, the reaction mixture was analyzed using RP-HPLC (Figure 1A, Figure 2A). The geS2U(34)–RNA^Lys^ product appeared as a single peak at Rt 19.07 min (Rt 14.32 min for the S2U(34)–RNA^Lys^ substrate). The samples collected during the RP-HPLC were analyzed using ESI-MS (Appendix A) and the proper molecular weight of the geS2U-RNA product was confirmed (MW 5485 g/mol and a measured mass of 5484.8 Da). The yield of the geranylation reaction, which was determined on the basis of comparing the areas under the peaks of the S2U–RNA substrate and the geS2U RNA product in the HPLC chromatogram, was estimated to be ~91%. 

The selenation of geS2U(34)–RNA^Lys^ produced a single product at Rt 14.1 min (Figure 1B, Figure 2B). The ESI-MS analysis confirmed the proper molecular weight of the Se2U RNA product (MW 5396 g/mol and a measured mass of 5395.58 Da) (Appendix A). The same geranylation and selenation reactions of 17-mer ASL, in which 2-thiouridine was substituted in the C5 position of uracil ring by the methylaminomethyl- group (mnm5S2U-RNA^Lys^), produced similar yields (geranylation > 90% and quantitative selenation) (Appendix A).

MBP–SelU, as with SelU–His_6_, requires the presence of Mg^2+^ ions for its enzymatic activity [11,20] and a concentration of 10 mM was considered optimal for achieving the maximum enzymatic efficiency of MBP–SelU. The application of MgCl_2_ in a higher concentration decreased the enzymatic activity of MBP–SelU to 73% for 25 mM MgCl_2_ and 43% for 100 mM MgCl_2_.

A comparison of the kinetic data, which were obtained for both reactions catalyzed by the MBP–SelU enzyme (Appendix A), to the previously obtained and already published kinetic constants for SelU–His_6_ [11], cited in the UniProt database www.uniprot.org/uniprot, (accessed on 23 February 2022; access NO. P33667), clearly indicated the significant increase in the enzymatic efficiency of the MBP–SelU enzyme. The geranylation and selenation reactions catalyzed by MBP–SelU were 1000- and 50-times more effective than reactions catalyzed by SelU–His_6_, respectively (Appendix A). Despite the fact that both reactions (with MBP–SelU or with SelU–His_6_) were catalyzed by the same enzyme (SelU) associated with the specific endogenous tRNAs, the MBP tag in MBP–SelU fusion construct seemed to act as a “molecular chaperone”, which significantly stabilized the recombinant protein and caused a significant increase in its enzymatic activity [30]. This high stability assured the good performance of all of the designed experiments for the enzyme substrate specificity. 

### 3.3. MBP–SelU Recognizes the Position of S2U in the RNA Chain, the Sequence Flanking S2U, and the Length and Structure of the RNA Substrate

The position of the S2U modification in the RNA chain, as well as the S2U flanking sequence and the length of the oligonucleotide substrate, has a significant impact on the recognition by the enzyme and, ultimately, on the efficiency of the MBP–SelU synthase, especially during the geranylation reaction. Table 1 and Appendix A show the structures, sequences, and lengths of the oligo-RNA substrates that were used in this study and their influence on the efficiency of the reactions catalyzed by the MBP–SelU enzyme. The geranylation and selenation reactions were performed under the conditions described in Appendix A. The results of the reactions are shown in the HPLC chromatograms and Appendix A and are summarized in Table 1.

The MBP–SelU enzyme showed the highest geranylation efficiency (~91%) for the S2U(34)–RNA^Lys^ 17-mer substrate, which most closely mimicked the native tRNA^Lys^ anticodon stem-loop and contained the 7-nt loop sequence: 5′-CUS2UUUAA-3′. Any shift of S2U in the ASL loop structure resulted in poorer recognition by the enzyme and impaired the efficiency of the geranylation reaction. Shifting S2U one position upstream (toward the 5′ end), as in S2U(33)–RNA (7-nt loop sequence: 5′-CS2UUUUAA-3′), resulted in a 10-fold decrease in geranylation efficiency (up to 9%). When S2U was placed downstream (toward the 3′ end) into the 35th position (7-nt loop sequence: 5′CUUS2UUAA-3′) or the 36th position (7-nt loop sequence: 5′-CUUUS2UAA-3′), the yield of the geranylation reaction decreased to 65% and 13%, respectively. A slight change in the loop sequence, despite the presence of S2U in the 34th position, as in the ASL–RNA^Arg^, mimicking the natural tRNA^Arg^, in which S2U was inserted instead of R5U (7-nt loop sequence: 5′-CUS2UCUAA-3′), significantly decreased the geranylation efficiency up to 10%. The obtained results indicated that the S2U modification specifically defined the loop structure ASL–RNA and that any positional shift of the S2U modification altered the ASL interaction with the SelU protein.

The S2U(34)–RNA^Lys^ and S2U(34)–RNA^Arg^ differed in the sequence of the stem and in the type of downstream nucleoside in the 35th position (U in S2U(34)–RNA^Lys^ and C in S2U(34)–RNA^Arg^) (Appendix A). Replacing U35 with C35 in S2U(34)–RNA^Lys^ decreased the geranylation yield 15-fold (91% and ~6%, respectively), whereas the ASL–RNA^Lys^ substrates that contained purines in the 35th position (A or G) were barely recognized by the enzyme, i.e., the products of the geranylation reaction occurred at very low levels (0.3–3%) (Table 2).

We also determined the length of the RNA substrate, which could be recognized by the enzyme. The 17-mers that mimicked the structure of ASL–tRNA^Lys^ were good substrates for the MBP–SelU synthase that recognized the oligonucleotide substrate and catalyzed the geranylation and selenation reactions with the high yield (91% and 100%, respectively). When the oligonucleotide was truncated to the 7-mer (tRNA^Lys^ anticodon loop model, AL) or 3-mer (tRNA^Lys^ anticodon model, A), the yield of the geranylation reaction decreased to 25% and 14%, respectively, which indicated that the substrate recognition by the enzyme was disabled. This suggested that the enzyme required the fully structured loop to exert its catalytic activity. The single nucleosides (S2U, mnm5S2U, and geS2U) did not serve as substrates for the enzyme (Table 1).

### 3.4. Monitoring the Relative Binding Affinity of the MBP–SelU Protein for Modified RNAs Using Microscale Thermophoresis (MST)

The microscale thermophoresis (MST) experiments were performed to investigate the relative binding affinity of the MBP–SelU protein to model ASL–RNAs (17-mers) (Table 3 and Appendix A), which differed only in the single nucleoside modification (U-, S2U-, geS2U- or Se2U-) at the site that corresponded to the wobble position in tRNA^Lys^. ASL–RNAs were labeled with Cy3 dye at the 5’ end. All experiments were performed according to the procedure described in the Materials and Methods section. The fluorescence intensity of the Cy3-RNAs was determined using the Capillary Scan method, and set to 300 counts. The initial MBP–SelU protein concentration determined using the Bradford protein assay was 452 µM. The samples contained the Cy3-labeled oligonucleotide and MBP–SelU protein in serial dilutions were incubated for 30 min, centrifuged, poured into standard MST capillaries and analyzed. The observed ligand-dependent changes in the initial fluorescence increased with the increasing concentrations of the protein (Appendix A). In the typical MST experiment, the fluorescence values of individual samples should not differ by more than 10%, but we observed a 2–3-fold change in the fluorescence intensity of the samples. The loss of material due to non-specific adsorption on the capillary walls or aggregation was excluded in the SD-test, which was recommended by the manufacturer in order to distinguish between fluorescence changes caused by the specific interactions between the target and ligand and non-specific effects. The SD-test confirmed that all measured protein–RNA interactions were specific. 

Due to the ligand concentration-dependent changes in fluorescence (>±10%), the Initial Fluorescence analysis, the method recommended in this case, was chosen in the MO Affinity Analysis software and the dissociation constant (K_d_) for each target–ligand pair was determined using the K_d_ fitting model. The changes in the initial fluorescence were represented as changes in raw fluorescence (ΔRaw Fluorescence) and plotted as a function of ligand concentration (Figure 3). Following the principles of the MST method [37], the calculated K_d_ confidence for the estimated K_d_ values of each pair that was studied was less than 68%, which indicated that the K_d_ values were within the specified range.

The results of the MST analysis, which are shown in Figure 3 and Table 3, as well as the additional information from the MST report (summarized in Appendix A), clearly confirmed that the SelU protein bound specifically all RNA tested oligonucleotides in contrast to the free MBP protein, which served here as the negative control and did not bind any of the studied RNAs. The differences in the relative affinity of SelU to the studied RNAs were small but significant. The affinity between SelU and the S2U- (studied interaction between enzyme–substrate) or Se2U-modified RNAs (studied interaction between enzyme–final product) was similar, although a slightly stronger interaction was observed between SelU and the thiouridine nucleoside than between SelU and the selenouridine-containing RNA (ΔK_d_ = −4.7 µM) (Table 3). The strongest interaction was observed between SelU and the *S*-geranyl RNA with K_d_ 3.95 µM. 

### 3.5. Determination of MBP–SelU Bound tRNA Content 

Consistent with the observations of Wolfe and Bjork performed on the wild-type or mutants of SelU synthase [23,24], the other SelU variants, such as MBP–SelU, the previously described SelU–His_6_, or the other tagged variants of SelU that were produced during our study (data not shown), also contained the tightly associated tRNA fractions. These observations indicated that SelU is an enzyme for which tRNA binding is an intrinsic property and that the tRNA–SelU interaction is not dependent on the attached tag. Further confirmation that tags are not involved in tRNA binding was the negative result of isolating tRNA from the pure MBP protein (0 µg of isolated tRNA), according to the procedure described in the Materials and Methods section. The similar isolation of tRNA from the pure MBP–SelU produced the positive result, i.e., a pool of tRNAs at the level of approximately one tRNA molecule per one MBP–SelU protein molecule. For this reason, all SelU variants had the unusual absorption spectrum, with the maximum at 260 nm, as for nucleic acids, and no peak at 280 nm, which is characteristic for proteins. The tRNA fraction bound to the SelU was easily observed in an ethidium bromide-stained SDS gel and in agarose gel (Appendix A). A northern blot analysis performed with specific DNA probes, which were complementary to the anticodon stem-loop domain of the bacterial tRNAs (tRNA^Glu^, tRNA^Lys^, and tRNA^Gln^) confirmed that these three types of tRNAs were associated with the MBP–SelU protein (data not shown). Our observations were consistent with those in the available databases, in which tRNAs specific to Lys, Glu, and Gln are described as the only ones that contain (c)mnm5S2U, (c)mnm5geS2U, and (c)mnm5Se2U modified nucleosiedes in the anticodon loop [5,6,38].

The UPLC-PDA-ESI(-)-MS analysis of the tRNAs that were isolated from the pure sample of MBP–SelU identified the number of high molecular weight compounds with molecular weights (MW) above 24 kDa, which corresponded to the MW of putative tRNA molecules. Based on the data from tRNAdb [38], we selected the types of bacterial tRNAs that could be substrates for SelU synthase. All names and sequences are listed in Appendix A. Each of these tRNAs could contain the following modifications in the wobble position, depending on the growth conditions of the bacterial cells: U, nm5U, mnm5U, cmnm5U, S2U, nm5S2U, mnm5S2U, cmnm5S2U, geS2U, nm5geS2U, mnm5geS2U, cmnm5geS2U, Se2U, nm5Se2U, mnm5Se2U, cmnm5Se2U. We calculated the atomic composition and molecular weight of all possible tRNA variants (Appendix A) and used these data to identify the tRNAs that interacted with SelU. Figure 4 shows the PDA chromatogram and deconvoluted mass spectra obtained from the UPLC-PDA-ESI(-)-MS analysis of the full-length tRNAs that were associated with MBP–SelU. Table 4 summarizes the identified and unidentified tRNAs whose masses were detected in the UPLC-PDA-ESI(-)-MS analysis.

The last (right-hand) column in Table 4, denoted “After Selenation”, shows the masses of the tRNAs with the R5Se2U modification, which were obtained after the addition of the portion of SePO_3_^3^^−^ into the MBP–SelU sample and 30 min incubation. During this time, the *S*-geranylated tRNA molecules underwent the selenation reaction (i.e., the *S*-geranyl group in the form of thiogeraniol (MW 170.31 g/mol) was replaced by selenium (MW 78.96 g/mol) (ΔMW = 91.35 g/mol)). This was further confirmation that the MBP–SelU contained mainly *S*-geranyl-substituted tRNAs that were selenated after the addition of SePO_3_^3−^. Some of the peaks in the deconvoluted mass spectra resulted from the UPLC-PDA-ESI(-)-MS analysis remained unidentified and are listed in the bottom panel of Table 4. We were unable to assign them to the specific tRNA using the MW calculated from a full sequence from tRNAdb (Appendix A). However, all of these tRNAs changed from the *S*-geranyl derivative to the selenium derivative after selenation in the protein, which was similar to the identified tRNAs that are shown in the top panel of the Table 4.

### 3.6. Identification of the Patterns of Nucleoside Modification in tRNAs Bound to the MBP–SelU Protein

The MBP–SelU-associated tRNAs, which were isolated from the pure protein preparations using phenol–chloroform extraction (see Materials and Methods section), were hydrolyzed using two nucleases (Benzonase and Phosphodiesterase I) and then dephosphorylated using Alkaline phosphatase to obtain the nucleoside mixture [39,40]. It was known from previous experiences, as well as from the literature [10,23], that the presence of the geranyl group inhibits the hydrolysis of the adjacent phosphodiester bond, which leads to the formation of 5′-R5geS2UpU-3′ dimers. The use of a prolonged reaction time and increased amounts of enzymes led us to the formation of the corresponding monomers (97%), but residual dimers (~3–8%) were still observed in three R5-substituted derivatives (cmnm5geS2UpU, mnm5geS2UpU, and nm5geS2UpU). Moreover, we found that the hydrolysis of the aforementioned dimers resulted not only in the formation of the correct monomers (cmnm5geS2U, mnm5geS2U, and nm5geS2U), but also in the higher molecular weight products of X-cmnm5geS2U, X-mnm5geS2U, and X-nm5geS2U (*m/z* of native nucleoside +154 Da). The dimers and non-specific hydrolysis products (X) were not observed for geS2U, any of the identified R5Us or for other modified or canonical nucleosides.

Unfortunately, the presence of dimers and products of non-specific hydrolysis (X) did not allow for the quantification of the identified modified nucleosides, so we only performed a qualitative assessment. In addition, we confirmed that the tested nucleosides were not decomposed during the hydrolysis reaction (or the decomposition could be excluded due to the low percentage (<2%)) and that the conditions used were safe for all of the tested compounds.

The nucleoside mixture that was obtained after tRNA hydrolysis was subjected to UPLC-PDA-ESI(-)-HRMS analysis and compared to the corresponding nucleoside standards (Figure 5). The list of nucleoside standards that were synthesized in the project [14,17,31,32,33] and used in the study, their molecular formulae, and UV characteristics can be found in the Appendix A. Figure 5A shows the chromatogram of the extracted ions (XIC) of the mixture of fifteen nucleoside standards obtained through the UPLC-PDA-ESI(-)-HRMS analysis. Equal amounts of nucleoside standards were used to prepare the mixture. The differences in the sizes of the mass peaks (the areas under the peaks) were due to differences in the degree of ionization of the compounds. Overall, 2-thiouridines (R5S2Us) and 2-selenouridines (R5Se2Us) were not detected in the hydrolysate of tRNAs associated with SelU. In contrast, these tRNAs were enriched in nucleosides that mainly contained the geranyl group, including geS2U, mnm5geS2U, cmnm5geS2U, and nm5geS2U (Figure 5B, Appendix A). Thus, the SelU protein showed the highest affinity for tRNAs with geranyl group modification. In addition, the C5-substituted uridines, including mnm5U, cmnm5U, and nm5U, were detected in lower amounts. Other modifications present in tRNA^Lys^, tRNA^Glu^, and tRNA^Gln^ were also identified (Appendix A). In the absence of suitable standards, this identification was based on their molecular weight only.

### 3.7. SelU Synthase Catalyzes the Conversion of Bacterial R5S2U-tRNA into R5Se2U-tRNA via the R5geS2U-tRNA Intermediate Product

The pure tRNA fraction was extracted from *E. coli* BW25113 SelU knockout strain (Δ*(araD-araB)567,* Δ*lacZ4787(::*rrnB-3*),* Δ*ybbB786::*kan*, λ-, rph-1,* Δ*(rhaD-rhaB)568, hsdR514*) using biotin-containing specific probes, as described in the Materials and Methods section. The UPLC-PDA-ESI(-)-HRMS analysis of the nucleoside composition of the isolated tRNAs (tRNA^Lys^, tRNA^Glu^, and tRNA^Gln^) confirmed that they mainly contained the R5S2U modification (99.05%) and the residual portion of R5U (0.95%) (data not shown). The R5geS2U and R5Se2U nucleosides were not found in the tested samples.

The R5S2U-tRNA^Gln^ substrate (as well as the R5S2U-tRNA^Glu^, data not shown) was used for SelU-catalyzed reactions, according to the scheme that is shown in Figure 6A: (i) geranylation through the incubation of the tRNA substrate with GePP and the MBP–SelU enzyme; (ii) geranylation and subsequent selenation through the incubation of the tRNA substrate with GePP and the MBP–SelU enzyme and then with SePO_3_^3^^−^ and the MBP–SelU enzyme; (iii) direct selenation through the incubation of the tRNA substrate with SePO_3_^3^^−^ and the MBP–SelU enzyme. The detailed procedure for each reaction is described in the Materials and Methods section. The R5S2U-tRNA^Gln^ substrate dissolved in the reaction buffer only, without the enzyme was used as a control for reactions i, ii, and iii. The qualitative analysis of the full-length tRNA reaction products was performed using UPLC-PDA-ESI(-)-MS. The analysis of the peaks in both the UV (for Rt determination) and XIC chromatograms (for MW determination) led us to the following conclusions. The geranylation of the R5S2U-tRNA substrate (R5S2U-tRNA→R5geS2U-tRNA) proceeded with high efficiency. Almost all of the tRNA substrate (Rt_(R5S2U-tRNA)_ 10.2 min) was converted into the *S*-geranyl product R5geS2U-tRNA (Rt_(R5geS2U-tRNA)_ 18.9 min) with an MW difference (ΔMW_1_, meant as MW_(R5geS2U-tRNA)_ − MW_(R5S2U-tRNA)_) that was assessed as 136 Da (Figure 6B). The geranylation of R5S2U-tRNA followed by selenation (R5S2U-tRNA→R5geS2U-tRNA→R5Se2U-tRNA) resulted in the selenated product R5Se2U-tRNA. We observed a shift in retention time from Rt_(R5S2U-tRNA)_ 10.2 min, which was characteristic of the reaction substrate (R5S2U-tRNA), to Rt_(R5geS2U-tRNA)_ 18.9 min characteristic of the first product (R5geS2U-tRNA), and then to (Rt_(R5Se2U-tRNA)_ 6.14 min) characteristic of the final product (R5Se2U-tRNA). The obtained R5Se2U-tRNA products had a different molecular weight from the R5S2U-tRNA substrate (ΔMW_3_ as MW_(R5Se2U-tRNA)_ − MW_(R5S2U-tRNA)_ was 47 Da) and the R5geS2U-tRNA product (ΔMW_2_ as MW_(R5Se2U-tRNA)_ − MW_(R5geS2U-tRNA)_ was −89 Da) (Figure 6B). No products of the direct selenation of the initial substrate (R5S2U-tRNA→R5Se2U-tRNA) were observed, which was similar to the model ASL–RNA. The same procedure repeated for the R5S2U-tRNA^Glu^ substrate and the R5S2U-tRNA^Lys^ substrate (data not shown) resulted similar effects, which confirmed the assumed course of the reactions.

In addition, the obtained data were confirmed by the analysis of the nucleoside composition of the products of the R5S2U-tRNA^Lys^ geranylation and selenation. For this, the samples after each reaction were hydrolyzed by two nucleases (Benzonase and snake venom Phosphodiesterase 1, svPDE 1) and dephosphorylated by Alkaline phosphatase, as described in the Materials and Methods section, and the resulting nucleoside mixtures were analyzed using UPLC-PDA-ESI(-)-HRMS. The results are shown in Figure 6C and Appendix A. Similar to the previous analysis of R5geS2U-tRNA hydrolysis products, we observed the formation of modified nucleoside monomers (R5S2U, R5geS2U, R5Se2U, and R5U, depending on the reaction) and the residue (<5%) of nucleotide dimers (R5S2UpU, R5geS2UpU, and R5Se2UpU). Dimers for R5UpU were not found. Again, we observed the appearance of native products containing the X group (*m*/*z* of native nucleoside + 154 Da), which resulted from the non-specific activity of the nucleases during dimer cleavage. The products with the X group were formed not only during the hydrolysis of the R5geS2UpU dimers, but they were also present in all of the samples that were analyzed (X-mnm5S2U, X-mnm5geS2U, and X-mnm5Se2U). The presence of dimers and products of non-specific hydrolysis did not allow for the quantification of the identified modified nucleosides, so we only performed a qualitative assessment (Figure 6C). The analysis of reaction products performed on the bacterial tRNAs is further evidence that the conversion of R5S2U-tRNA into R5Se2U-tRNA is a two-step process and that the R5geS2U-tRNA that is formed in the first step is the intermediary in the R5Se2U-tRNA formation process.

## 4. Discussion

In the study presented here, we discussed the requirements for the substrates of the tRNA 2-selenouridine synthase (SelU), the composition of the tRNA molecules bound to this nucleoprotein, in particular the pattern of nucleoside modifications, and we provided unequivocal evidence that confirmed the postulated cellular pathway of the R5S2U→R5Se2U transformation in bacterial tRNAs. In the presented study, we used the fusion protein with an MBP tag at the N-terminus of SelU and proved that the presence of the MBP tag significantly stabilized the SelU protein, as reflected by its high activity in the specific reactions of geranylation and selenation. We do not know the mechanism through which the MBP interacted with the SelU. At this stage, we can only speculate that the MBP appeared to act as a “molecular chaperone”, which promoted the stability of the proteins that were fused to it, and that its maintenance function likely relied on direct stabilizing contact with the attached protein [30,41]. The catalytic efficiency of the studied MBP–SelU variant was assessed, under conditions that were described previously [11], as being 1000-fold higher than that of SelU–His_6_ during geranylation and 50-fold higher than that during selenation (Appendix A, Appendix A). The increased stability improved the catalytic performance of the enzyme, which could be consistent with the stability and efficacy of SelU under native conditions, i.e., in bacterial cells. Thanks to these parameters of MBP–SelU, we had the necessary molecular tool to characterize and evaluate the properties of this enzyme.

We found that SelU recognized the position of the S2U modification in the anticodon loop, the S2U flanking sequence, and the length of the RNA substrate. The enzyme recognized the simplest chemically synthesized ASL–RNA model substrate (17-mer), which mimicked the native anticodon loop of tRNA^Lys^ with a 7-nt loop sequence (5′-CUS2UUUAA-3′) during the geranylation and selenation reactions and achieved the high efficiency in the applied conditions (4.5-fold excess of ASL–RNA substrate for 1 h of incubation at 37 °C). Shifting the S2U position in the ASL loop structure (to the 33^rd^, 35^th^ or 36^th^ position) resulted in a much poorer recognition by the enzyme and impaired the efficiency of the geranylation reaction (Table 1). S2U(34)–RNA substrates that contained purine residues A or G in the 35^th^ position were barely accepted by the enzyme, whereas natural U(35) was the better substrate compared to the C(35) variant (Table 2). The length of the oligonucleotide substrate that mimicked the structural elements of tRNA^Lys^ also affected recognition by the enzyme. The oligonucleotide that truncated to 7-nt, which mimicked the anticodon loop (AL) of tRNA^Lys^, and the oligonucleotide truncated to 3-nt, which mimicked only the anticodon (A), were less accepted and the efficiency of the enzyme in the geranylation reaction decreased to 25% and 14%, respectively. This indicated that the enzyme required the fully structured loop to exert its catalytic activity. The single nucleosides (S2U, mnm5S2U, and geS2U) did not serve as substrates for the MBP–SelU enzyme. Interestingly, the enzyme was less demanding in terms of substrate specificity during the selenation reaction, as all geS2U–ASL models tested were converted quantitatively (geS2U in the 33rd, 34th, and 35th position of ASL) or still produced the high yield (66% for geS2U in the 36^th^ position of ASL) for their selenouridine analogs. The MBP–SelU enzyme also accepted the geS2U–RNA^Arg^ ASL in the selenation reaction, which proceeded with a 94% yield. These results suggested that the recognition of tRNA substrates by the enzyme occurred through different mechanisms in the two reactions that were tested (i.e., alkylation at the sulfur atom in geranylation and nucleophilic substitution of selenophosphate at C2 in selenation) [11].

MBP–SelU and other fusion variants that were constructed in our studies (data not shown) had the tightly bound tRNA fraction that altered the spectral properties of the protein, which had an unusual absorption spectrum with a maximum at 260 nm, as in nucleic acids, and no peak at 280 nm, as in proteins. We roughly assessed the amount of tRNA associated with MBP–SelU as approximately one tRNA molecule per one protein molecule after the isolation of tRNA from the pure protein using the phenol–chloroform extraction method. Additionally, using the northern blot analysis, we found that three types of tRNAs (tRNA^Glu^, tRNA^Lys^, and tRNA^Gln^) were associated with the MBP–SelU protein (data not shown). Then, using the UPLC-PDA-ESI(-)-MS technique, we identified the full-length tRNAs associated with the MBP–SelU protein (the results of the tRNA analysis are published for the first time) and assessed their nucleoside composition (Figure 4 and Figure 5, Table 4, Appendix A). The tRNAs that were associated with the MBP–SelU protein (exactly with SelU) were enriched in nucleosides that mainly contained the geranyl group (mnm5geS2U, geS2U, cmnm5geS2U, and nm5geS2U). The relatively high proportion of “naked” geS2U, without mnm- and cmnm- substituents in the C5 position, suggested that the biosynthesis of the final R5Se2U–tRNA occurred independently of the 5-substitution. Importantly, sulfur- and selenium-modified nucleosides were not detected in the hydrolysates of the tRNAs that were associated with the SelU protein. This result was confirmed by the microscale thermophoresis (MST) measurements, in which we assessed the relative binding affinity between the MBP–SelU protein and the synthetic RNAs, which contained a single modified uridine (S2U-, geS2U- or Se2U-) in the position that mimicked the wobble position in tRNA. Our results indicated that the geranylated RNA (Cy3–geS2U(34)–RNA^Lys^) had the highest affinity for MBP–SelU (K_d_ = 3.95 µM), whereas the other ASL–RNAs with 2-thio- and 2-seleno- modification bound to the enzyme in a similar range with a lower affinity. The Bjork group drew similar conclusions already in 2016, when they examined the modifications of tRNAs associated with wild type proteins, as well as mutants with the increased geranylation capacity, e.g., MnmH(G67E) [23], and he was able to identify the cmnm5geS2UpU, mnm5geS2UpU, and nm5geS2UpU dimers. They concluded that the tRNAs bound to the enzymes were enriched in the geranylated derivatives and suggested that the MnmH type proteins appeared to have the strong affinity for such tRNAs. In the first report on the R5geS2U–tRNA, Dumelin et al. suggested that the geranylation of tRNA impaired cellular translation efficiency and was involved in the readout of GAG codons that were specific to Glu or AAG encoding Lys [10]. If *S*-geranyl–tRNA performed this function, it should be available to the translation apparatus in the cytoplasm of the bacterial cell. In both, this study (data not shown) and Bjork’s study, no free R5geS2U–tRNAs were detected in the cytoplasm [23]. In our opinion, the total R5geS2U–tRNA is tightly associated with the SelU protein, which releases the final transformation product, i.e., the R5Se2U–tRNA, after the selenation of the R5geS2U–tRNA.

Finally, we demonstrated for the first time the pathway of conversion of native bacterial R5S2U–tRNA into R5Se2U–tRNA as the two-step process with the formation of R5geS2U–tRNA as the intermediate product, which is strongly associated with the SelU enzyme and plausibly is not involved in mRNA translation because the R5geS2U–tRNA content in the cellular fraction is negligible (data not shown). The reaction of direct R5S2U–tRNA selenation did not result in the R5Se2U–tRNA product.

## Data Availability

All relevant data are within the paper and Appendix A.

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
