# Peer review of "Escherichia coli tRNA 2-Selenouridine Synthase (SelU): Elucidation of Substrate Specificity to Understand the Role of S-Geranyl-tRNA in the Conversion of 2-Thio- into 2-Selenouridines in Bacterial tRNA"

_cells, 2022, doi:10.3390/cells11091522_

Round 1

Reviewer 1 Report

I don't find any problem. The manuscript can be published.

Author Response

Thank you very much for positive evaluation of our manuscript.

Reviewer 2 Report

The work of Szczupak and colleagues is a continuation of the previous works by this research group on the same topic. Compared with the previous studies, they used a different tag for SelU protein that allows the stabilisation of such recombinant protein and facilitates the in vitro analysis of this enzyme.

The experiments are performed correctly, they are scientifically sound, and the conclusions correspond to the results. However, the quality of the presentation must be improved taking into account the remarks below. 

  1. Line 73: Scheme 1 should be Figure 1.
  2. Line 101: nowadays S. typhimurium is called S.enterica serovar Typhimurium. Please mention this.
  3. Line 136: statement ..."whose gene was isolated from the E. coli RNA transcript"...is unclear. Was PCR of genomic DNA done or reverse transcription used to clone this gene? Please rewrite.
  4. Line 142: "compounds studied" should be replaced by "substrates studied".
  5. Line 254: "phenol solution" should be "phenol water solution"?
  6. Line 264:  "Abs at λ=260 nm" could be replaced by "A260".
  7. Lines 288/298: are UPLC-PDA-MS and UPLC-PDA-ESI(-)-HRMS different analysis methods and were used as described?
  8. Line 302: "For each type tRNA" should be "For each type of tRNA" or "For each tRNA type".
  9. Line 394: What does it mean "The ESI-MS analysis confirmed the structure of Se2U-RNA"? Is this confirmation of sequence including the presence of Se2U in tRNA as determined by MS? Please rephrase. 
  10. Line 409: are the numbers correct? Did you take into account that modifications may change the extinction coefficient of absorption of respective nucleoside? If this is not the case, please state it somewhere in the text.
  11. Line 420: Sentence "This high stability assures to easily perform all the designed"... seems not to be correct. Please rephrase to "This high stability assures the easy performance of all the designed"... or so.
  12. Line 434: Table 1 is not visible well, at least in the pdf format. Please assure that the pictures in it and elsewhere are of good quality.
  13. Line 654: ΔybbB786::kan should called selU::kan?
  14. Line 655: "Me-thods" should be "Methods".
  15. Line 795: "Bjork" should be referred to "Bjork and colleagues" or Bjork group (first author is Jäger).  

Author Response

Thank you for the positive review of our manuscript. We took into account all comments, which resulted in changes in the text of the manuscript. All changes in the revised manuscript are highlighted in red. Below is written our reply to individual questions and suggestions.

  1. Line 73: Scheme 1 should be Figure 1.

Scheme 1 has been renamed to Figure 1 and the remaining Figures have been renumbered.

  1. Line 101: nowadays typhimurium is called S.enterica serovar Typhimurium. Please mention this.

Now this sentence is: The enzymatic activity in extracts of Salmonella enterica serovar Typhimurium that catalyzes…

  1. Line 136: statement ..."whose gene was isolated from the coli RNA transcript"...is unclear. Was PCR of genomic DNA done or reverse transcription used to clone this gene? Please rewrite.

Now this sentence is: „In previous studies, we used the wild type SelU protein, whose gene was isolated from the E. coli RNA by reverse transcription followed by PCR with a specific primers, and cloned into the DNA expression plasmid pET28c…..”

  1. Line 142: "compounds studied" should be replaced by "substrates studied". Changed
  2. Line 254: "phenol solution" should be "phenol water solution"?

The name “phenol solution” is the commercial name of the reagent from Sigma Aldrich, however I changed it to a more specific term: “phenol equilibrated with 10 mM Tris HCl, pH 7.4 solution (Sigma-Aldrich)”

  1. Line 264:  "Abs at λ=260 nm" could be replaced by "A260". Changed
  2. Lines 288/298: are UPLC-PDA-MS and UPLC-PDA-ESI(-)-HRMS different analysis methods and were used as described?

Yes, both methods have been used. UPLC-PDA-ESI(-)-MS for determination of the high molecular weight compounds, like the RNA oligonucleotides or tRNAs, and UPLC-PDA-ESI(-)-HRMS for nucleosides. Both methods are described in detail in point 2.4.5 of Materials and Methods. The UPLC-PDA-ESI(-)-HRMS method allows to determine the molecular weight with high accuracy (up to the fourth decimal place) but is applicable to low molecular weight compounds with a mass below 1000.

  1. Line 302: "For each type tRNA" should be "For each type of tRNA" or "For each tRNA type". Changed
  2. Line 394: What does it mean "The ESI-MS analysis confirmed the structure of Se2U-RNA"? Is this confirmation of sequence including the presence of Se2U in tRNA as determined by MS? Please rephrase.

Of course, the structure of the reaction product was not determined, we confirmed only the expected molecular weight, which proves the correct course of the reaction. The sentence was rephrased to: „The ESI-MS analysis confirmed the proper molecular weight of Se2U-RNA product (MW 5396 g/mol, measured mass 5395.58 Da)…”

  1. Line 409: are the numbers correct? Did you take into account that modifications may change the extinction coefficient of absorption of respective nucleoside? If this is not the case, please state it somewhere in the text.

The efficiency of the reaction was assessed by HPLC analysis of the reaction mixture contents, (substrate and products), on the basis of the comparison of the areas under the peaks of the S2U(34)-RNA substrate and the geS2U-RNA product in the geranylation reaction, as well as for the geS2U-RNA substrate and the Se2U-RNA product in the selenation reaction. In the case of the second reaction (selenation), the determination of the yield of the reaction was simple because we observed the complete conversion of the geS2U-RNA substrate to the Se2U-RNA product.

You are absolutely right that the amount of substrate or product is affected by the extinction coefficient of the oligonucleotide, which is closely related to the extinction coefficient of the modified nucleoside. The extinction coefficients for S2U or Se2U are known (9.160 and 2.687 at 260 nm, respectively, Leszczynska et al. Int. J. Mol. Sci. 2020, 21, 2882). and were used when we counted the concentrations of the corresponding oligonucleotides. Unfortunately, the geS2U extinction coefficient is still unknown. Therefore, when we determined the concentration of geS2U-RNA, the coefficient of S2U was taken into account. However, the influence of the oligonucleotide extinction coefficient on the final reaction efficiency is rather small, ~2% for significantly different extinction coefficients S2U and Se2U (9.16 vs. 2.686). I took the reviewer's comment into account by correction of the enzyme efficiency values in the reaction without specifying the decimal values.

  1. Line 420: Sentence "This high stability assures to easily perform all the designed"... seems not to be correct. Please rephrase to "This high stability assures the easy performance of all the designed"... or so.

The sentence was rephrased: „This high stability assures to easy performance of all the designed experiments of the enzyme substrate specificity”.

  1. Line 434: Table 1 is not visible well, at least in the pdf format. Please assure that the pictures in it and elsewhere are of good quality.

The table was prepared again. Current table:

Yield

[%]

Nucleoside

Anticodon

(A)

Anticodon-Loop

(AL)

Anticodon-Stem-Loop

(ASL)

3-mer

7-mer

17-mer

S2U

mnm5S2U

geS2U

Geranylation

0

-

14.0 ±0.7

25.0 ±4.4

9.0 ±0.9

91.0 ±4.7

65.0 ±4.1

13.0 ±2.0

10.0 ±1

Selenation

0

0

nd

nd

100

100

100

66.0 ±1

94 ±2.6

  1. Line 654: ΔybbB786::kan should called selU::kan?

Description of the bacterial strain E. coli DSelU [Δ(araD-araB)567, ΔlacZ4787(::rrnB-3), ΔybbB786::kan, λ-, rph-1, Δ(rhaD-rhaB)568, hsdR514] comes originally from its authors from E. coli Genetic Stock Center Yale College, US.

  1. Line 655: "Me-thods" should be "Methods". Changed
  2. Line 795: "Bjork" should be referred to "Bjork and colleagues" or Bjork group (first author is Jäger). Changed

This manuscript is a resubmission of an earlier submission. The following is a list of the peer review reports and author responses from that submission.

Round 1

Reviewer 1 Report

The present study by Szczupak et al. reports a biochemical characterization of the E. coli tRNA modification enzyme SelU that converts 2-thio to 2-selenouridines in the wobble position of tRNAs. The authors have already published in 2018 (Sierant et al. FEBS Lett 2018) that this modification occurs in two successive steps with the S-geranyl-2-thiouridine being the intermediate. Here, by using a MBP fusion protein (MBP-SelU) that provide a much higher stability to the recombinant protein, they confirm that the modification indeed occurs in two successive steps (2-thiouridine -> S-geranyl-2-thiouridine -> 2-selenouridine). By using several mini-substrate, they also provide some hints relative to the substrate requirements by SelU. Overall, the data are scientifically sound but the manuscript suffers from a poor writing that renders the reading difficult. In addition, some points deserve clarifications. Therefore, I do not recommend publication of this report in its present form in Cells and suggest some modifications to strengthen the study below.

Major points:

1) I would really recommend to re-write the entire manuscript in a more concise way. Some technical information are not necessarily needed in the result section, provided that sufficient information is given in the method section. Similarly, the discussion starts here by a long repetition of the results part and is poorly placed in context (for instance, but not limited to, l747-751 are repetition of l401-405; l761-783 are repetition of the result part). 

2) The fact that the authors have already published in 2018 that this modification occurs in two successive steps with the S-geranyl-2-thiouridine being the intermediate, could somewhat be detrimental to the present study. However, I believe this is mainly a problem of manuscript writing. Here the focus is overall put on this very same aspect, whereas the study would clearly benefit from highlighting more convincingly the new results provided in this new manuscript.

3) The authors overexpress and purify the MBP fusion protein (MBP-SelU) with tightly bound RNAs (some of which, if not all of which are tRNAs). I clearly appreciate the fact that the authors have characterized these co-purified RNAs as containing some specific tRNAs (part 3.5). However, I really request to determine and report (at least an estimate) the protein/RNA ratio of their purified recombinant MBP-SelU. Is this RNA contamination not a problem for the determination of kinetic parameters for the SelU enzyme? Can kcat and KM values be trusted if MBP-SelU is contaminated? Similarly, how a clean Kd can be determined if (part of) the protein is already associated with RNAs? There are methods to remove RNAs associated with RNA-binding proteins, from the simplest high-salt buffer purification (1 M  NaCl; Here purification is performed at maximum 200 mM salt, and the final Superdex column is performed at 25 mM salt), to more sophisticated methods with PEI specific nucleic acids precipitations (see Bou-Nader et al. STAR Protoc. 2022 or Lee BM et al. JMB 1998). The authors should try to purify the MBP-fusion protein free of RNA, report their attempts, and/or explain why this is not feasible.

Minor points:

4) the common abbreviations used in the community for modifications consists of lower-case letters, with superscript numbers s2U not S2U (see for instance Boccaletto et al. NAR 2018 and the MODOMICS database). Please correct throughout the manuscript and figures.

5) the term "mechanism" used in the abstract and the manuscript can be mis-leading. In their study, the authors do not "explain the final mechanism of selenation". They identify/confirm the different steps involved in the selenation. "Mechanism" is here clearly not appropriate, please change with terms like "the order of the steps" or similar terms. Please change in the entire manuscript.

6) introduction l37: remove the term "begins". The order of the maturation steps is not as described here, but intricate with certain chemical modifications that can occur before the end processing (see for instance Hopper et al. Cell 1978 and Hopper Genetics 2013, Barraud et al. IUMB Life 2019, Jackman eLS 2010 for reviews).

7) introduction l68: not metyl, methyl.

8) introduction l137-139: "Although we worked with the bacterial gene, production of the recombinant SelU in the bacterial expression system was very ineffective, probably because the function of SelU synthase and its highly regulated expression in bacterial cells." Please rephrase and explain the meaning better.

9) Table 2, l440: "nd": please define. If not determined. Please explain why it could not be determined.

10) l446: "We also determined the minimum length requirements of the enzyme for the RNA substrate." This sentence is misleading. The authors have shown that the enzyme is active on the 17-mer stem-loop, but less active (for the geranylation part) on 3-mer or 7-mer RNAs. But the minimal length requirement has not been determined. Please rephrased appropriately.

11) Table 4 and l527-533: "is slightly stronger", "with the weakest affinity". The differences in the values reported in Table 4 are not statistically significant (except for the 3.95 +- 0.4 uM). So do not comment on these "differences".

12) l567: "we identified the tRNAs interacting with the protein" not "we identified the tRNAs present in the protein were idrntified."

13) l662: Figure 5B,C not 7B,C.

14) l664-673: Where are the Rt and MW values coming from? Which Figure exactly? I did not find this information. Please either direct the reader to the correct Figure, and add a new Figure that correspond to these data.

15) Figure5 B and C: It is not clear to me why no signal is seen in the mass spectrum of the direct selenetion reaction (right spectrum). I understand there is no direct selenetion of the initial substrate, but why is this initial substrate not detected as initially (left spectrum)?

16) l762: "SelU detects the position of the S2U modification in the anticodon, the S2U-flanking sequence, and the size of the RNA substrate." Is it really the size? I would say it is rather the structure of the RNA stem-loop (or at least the loop structure). Even if the data presented in here are not clearly sufficient to appreciate the structural requirements, please rephrase to reflect more the tested RNAs, ie stem-loop structure and not purely the size.

17) l762-766: " It appears that the absence of the substituent at position 5 of the S2U or geS2U nucleobase or the absence of the other modifications that occur naturally in the anticodon domain of bacterial tRNALys (e.g. N6-threonylcarbamoyladenosine, t6A, at position 37 or pseudouridine at position 39) does not significantly affect the catalytic efficiency of the SelU protein." Clearly this cannot be said, it has not been tested. Here, without substituent at position 5 of s2U or ges2U or t6A at position 37, the enzyme is active. But it could really be that the enzyme is 10-50 times more active or less active with this modifications present. This should be properly tested if the authors want to discuss this point.

18) Figure S8: This figure is really of poor quality. The different lines/colors cannot be seen. And please add a caption to explain the meaning of these curves.

Reviewer 2 Report

This manuscript will update the authors' previous paper (in 2018) with new samples, new methods, and new data. By preparing the high quality SelU enzyme and the tRNALys/Gln/Glu molecules from SelU-free E. coli cells, the authors reconstituted the S2U to Se2U conversion reaction in vitro. They confirmed that the natural tRNA substrates (and their moieties) are optimal substrates of SelU, as expected. It is convincing that the  S-geranylated tRNA molecules are intermediate rather than alternative final products.

  Although this manuscript will interest people who engaged in SelU and tRNA, general readers might not be satisfied enough. First, it is not unexpected that the S2U to Se2U conversion must occur via the activation of the sulfur atom by attaching a good leaving group, as stated in the paper 2018. General readers are not familiar with a wrong hypothesis. Second, fusion of a difficult-to-express protein with a MBP tag is the first thing to do for in vitro enzyme assay. Because the main text of this manuscript is very precise and redundant, readers might expect a huge technical leap. Third, this manuscript provides no three-dimensional structural information about the enzymatic mechanism of SelU. Thus, readers can imagine only the before after of the reaction. I think that these three points can be improved in the revised manuscript. I also found a few typos.

  I suggest to add an additional experiment. As shown in the Scheme 1 of the paper 2018, HSGe is a byproduct of the SelU reaction. Is it possible to detect and determine the quantity of this molecule? I think that this would be the most direct evidence to prove the two-step mechanism of SelU.

  I am personally interested in any interaction of the SelU and SelD, because someone reported the interaction of SelD and SelA to directly transfer selenophosphate which may not be stable in the cytoplasm. Also, I don't know how GePP is safely transferred to SelU. How general is the SelU-SelD system in the bacterial kingdom? These points could be discussed.

Round 2

Reviewer 1 Report

Szczupak and coworkers have submitted a revised version of their initial manuscript. Although they address some of the points raised during the initial review, I am overall not satisfied at all with several responses and with the revised version and cannot therefore recommend this manuscript for publication.

Here are the main problematic answered to point initially raised. The authors had the chance to correct their manuscript following the recommendations of the first review round, but unfortunately did not seem to take this opportunity into consideration.

Main point #3:

The authors overexpress and purify the MBP fusion protein (MBP-SelU) with tightly bound tRNAs. Although they could not assess the protein/tRNA ratio of their true sample, they assume that ratio reported by others (Wolfe et al. 2004), namely 1 SelU protein for 2 tRNAs is also the one of their own sample after purification. The authors claimed that "The tRNA introduced into the reaction in this way was always treated as background and subtracted from the results obtained", there is no information in the manuscript "material and methods" section that would indicate how this was actually done. From the manuscript, it seems that the simple Michaelis-Menten model was assumed to determine the kcat and Km of SelU, although this model is not relevant for an enzyme which comes bound to tRNA molecules that need to dissociate first. Values of kcat and Km reported here are irrelevant.

In addition, values of dissociation constants of SelU with various stem-loop RNAs determined by microscale thermophoresis are also irrelevant. There is no information in the manuscript to explain how the bound fraction of tRNAs is taken into account in these binding experiments. The Kd values reported here depends on the dissociation constants of the initially bound tRNAs. The values reported here are clearly irrelevant and do not reflect the true binding event of the studied RNAs. 

Minor point #15:

In Figure 5 B and C, I initially requested an explanation that justified the fact that no signal is seen in the mass spectrum of the direct selenetion reaction (right spectrum). I can understand that there is no direct selenetion of the initial substrate, but why is this initial substrate not detected as initially (left spectrum)? The authors did not answer at all to my request (i.e. "Figure 5 was changed and completed"), but decided to simply remove this panel from the Figure. I find this situation very uncomfortable. To me, if the authors had the intention to hide something, they would not have acted differently. Not answering and removing the problematic figure is to me quite astonishing. Without a proper answer and without the raw data, I cannot judge whether this situation can be qualified as a fraud, but the way the authors have addressed this point is clearly problematic.

Minor point #17:

The authors do not want to reconsider their statement and still write: "It appears that the absence of the substituent at position 5 of the S2U or geS2U nucleobase or the absence of the other modifications that occur naturally in the anticodon domain of bacterial tRNALys (e.g. N6-threonylcarbamoyladenosine, t6A, at position 37 or pseudouridine at position 39) does not significantly affect the catalytic activity of the SelU protein, as the yields of the reactions on these minimal substrates are high or quantitative." But as mentioned in the initial review, this point has not been tested at all experimentally. I repeat, this should be properly tested if the authors want to discuss this point. Maybe they are right, but a plausible hypothesis is not the same as an experimentally validated point.

Reviewer 2 Report

The revised manuscript gave me an impression that each step was properly examined, confirmed, and discussed. The revised manuscript is not an update of the authors' previous studies but gives a final conclusion for this research theme.